# Improving Out-of-Distribution Detection with Markov Logic Networks

**Konstantin Kirchheim** [1]   **Frank Ortmeier** [1]

## Abstract

Out-of-distribution (OOD) detection is essential for ensuring the reliability of deep learning models operating in open-world scenarios. Current OOD detectors mainly rely on statistical models to identify unusual patterns in the latent representations of a deep neural network. This work proposes to augment existing OOD detectors with probabilistic reasoning, utilizing Markov logic networks (MLNs). MLNs connect first-order logic with probabilistic reasoning to assign probabilities to inputs based on weighted logical constraints defined over human-understandable concepts, which offers improved explainability. Through extensive experiments on multiple datasets, we demonstrate that MLNs can significantly enhance the performance of a wide range of existing OOD detectors while maintaining computational efficiency. Furthermore, we introduce a simple algorithm for learning logical constraints for OOD detection from a dataset and showcase its effectiveness.

## 1. Introduction

Deep Neural Networks (DNNs) (LeCun et al., 2015; Schmidhuber, 2015) provide state-of-the-art performance across various computer vision tasks, including image classification (Dosovitskiy et al., 2021), object detection (He et al., 2017), and semantic segmentation (Ronneberger et al., 2015). However, DNNs are prone to making incorrect predictions with high confidence when applied to data outside their training distribution (Nguyen et al., 2015). This property motivates out-of-distribution (OOD) detection mechanisms (Yang et al., 2024; Kirchheim et al., 2022), which aim to identify OOD inputs at runtime. In recent years, a wide array of OOD detectors has been developed, leveraging the representation of the input at different layers of a DNN, such as predicted posteriors (Hendrycks & Gimpel, 2017), unnormalized logits (Zhang et al., 2022), or deeper latent representations (Lee et al., 2018; Wang et al., 2022). These detectors often construct a model of the neural representations for in-distribution (ID) data and classify inputs with low probability under this model as OOD. While providing reasonable performance in many settings, current detection methods face several limitations. Firstly, they rely on potentially superficial statistical patterns in the representations learned by DNNs but neglect the semantics of these representations in the context of the task. Secondly, they lack explainability, as they generally provide only a scalar outlier score without further justification for why an input is marked as OOD. Thirdly, integrating prior knowledge about the structure of the training distribution into existing detectors remains challenging, as this knowledge often concerns abstract concepts that are difficult to correlate with high-dimensional inputs. For instance, many people would intuitively classify blue stop signs as OOD because stop signs are typically red. However, embedding such domain-specific knowledge into current OOD detectors is difficult due to their reliance on entangled, opaque neural representations.

Recently, neuro-symbolic approaches for OOD detection have shown promise in addressing these limitations. For example, LogicOOD (Kirchheim et al., 2024) detects human-understandable concepts in inputs and verifies their configuration against a set of logical constraints. While this method increases performance on some datasets and offers a degree of explainability, its reliance on strict logical rules can be too rigid for complex, real-world applications where probabilistic associations dominate.

This work proposes integrating existing OOD detectors with a probabilistic graphical model - specifically, a Markov logic network - defined over human-understandable concepts. Such a model offers explainability and allows the seamless integration of prior knowledge. In particular, we make the following contributions:

1. We introduce a novel OOD detection framework based on Markov logic networks (MLNs), leveraging probabilistic reasoning over logical constraints (Section 3).

2. We propose a novel algorithm to automatically learn human-interpretable constraints for OOD detection

[1]Department of Computer Science, University of Magdeburg, Germany. Correspondence to: Konstantin Kirchheim <kirchhei@ovgu.de>.

*Proceedings of the 42ⁿᵈ International Conference on Machine Learning*, Vancouver, Canada. PMLR 267, 2025. Copyright 2025 by the author(s).

from a dataset (Section 4).

3. We demonstrate that combining MLNs with existing OOD detectors significantly improves performance across several datasets, detectors, and DNNs (Section 5).

## 2. Background & Related Work

### 2.1. OOD Detection

Let $f : \mathcal{X} \to \mathbb{R}^N$ be a classifier that maps points from its input space $\mathcal{X} \subseteq \mathbb{R}^K$, drawn according to a distribution $p_{\text{data}}$, to a vector of class logits. An OOD detector for $f$ is a scoring function $D : \mathcal{X} \to \mathbb{R}$ that maps inputs to scalar outlier scores, such that OOD inputs, which have a low probability under $p_{\text{data}}$, receive a higher score than ID data points that have a high probability under $p_{\text{data}}$. A simple thresholding scheme then makes the final decision as:

$$\text{outlier}(\mathbf{x}) = \begin{cases} \texttt{true} & \text{if} \quad D(\mathbf{x}) \geq \tau \\ \texttt{false} & \text{else} \end{cases}. \quad (1)$$

In recent years, numerous OOD detection methods have been developed, with comprehensive overviews available in recent surveys (Lu et al., 2024; Yang et al., 2024; Kirchheim et al., 2022). Broadly, many of these methods can be categorized into the following groups:

**Posteriors** Class membership probabilities can be obtained from a classifier $f$ by applying the normalizing softmax function $\sigma$ to its output. The Maximum Softmax Probability (MSP) baseline method (Hendrycks & Gimpel, 2017) uses the negative maximum class posterior, $-\max_i \sigma(f(\mathbf{x}))_i$, as an outlier score. Ensembling (Lakshminarayanan et al., 2017) and Monte-Carlo Dropout (Gal & Ghahramani, 2016) can enhance the performance of this baseline approach at the expense of additional computation.

**Logits** It has been observed that the softmax-normalized DNN posteriors are systematically biased, and methods based on unnormalized logits can improve OOD detection. Consequently, approaches such as using the maximum logit (Hendrycks et al., 2022) or its smooth variant, the Energy-based score (EBO) (Liu et al., 2020), computed as $-\log \sum_i \exp(f(\mathbf{x})_i)$, have been proposed.

**Latent Representations** Typical DNN classifiers can be decomposed into a feature extractor $\Phi$ that maps inputs to latent representations, and a classification head $h$, such that $f = h \circ \Phi$. The Mahalanobis method (Lee et al., 2018) fits a multivariate Gaussian to the latent representations of each ID class and then uses the Mahalanobis distance of new data points as an outlier score. Simplified Hopfield Energy (SHE) (Zhang et al., 2022) learns a center $\mu_y$ for each ID class and uses $-\mu_y^\top \phi(\mathbf{x})$ as the outlier score, where $y$ is the maximum a posteriori class for $\mathbf{x}$.

**Latent Rectification** More recently, methods that rectify the representations of neural networks have been proposed. Mathematically, these methods introduce a rectifier $f = h \circ r \circ \phi$ that alters the latents. Examples include DICE (Sun & Li, 2022), which sparsifies the activations, as well as ReAct (Sun et al., 2021) and ASH (Djurisic et al., 2023), which replace unusual activations with alternative values. These modified representations often provide better performance when used with a downstream OOD detector.

### 2.2. First-Order Logic

First-order logic (FOL) is a formal system for reasoning about objects, their properties, and their relationships. In FOL, a domain $\mathcal{X}$ defines the set of objects being considered. Predicates $\mathcal{P}$ represent properties or relationships among objects, and functions $\mathcal{F}$ map objects to other objects in the domain. Such predicates could include, for example, $\texttt{RED} : \mathcal{X} \to \{\text{true}, \text{false}\}$, which supposedly evaluates to true for objects that are of red color. FOL formulas combine predicates and terms using logical connectives such as conjunction ($\land$), disjunction ($\lor$), negation ($\neg$), implication ($\to$), exclusive or ($\oplus$), and equality ($=$). Quantifiers specify whether a formula applies universally ($\forall$) or existentially ($\exists$) over the domain. The semantics of a FOL formula is provided by an interpretation $\mathcal{I}$ which maps predicates to functions $\mathcal{P}^{\mathcal{I}} : \mathcal{X}^n \to \{\text{true}, \text{false}\}$, determining which tuples of objects satisfy the predicate, and functions $\mathcal{F}$ to functions $\mathcal{F}^{\mathcal{I}} : \mathcal{X}^n \to \mathcal{X}$, defining their behavior over the domain.

As we only consider single inputs in this work, we will use a subset of FOL consisting of universally quantified statements over unary predicates and functions in the following.

### 2.3. Markov Logic Networks

Markov logic networks (MLNs) (Richardson & Domingos, 2006) constitute a probabilistic generalization of first-order logic (FOL) and are instances of the neuro-symbolic computing paradigm (Besold et al., 2021). They can also be seen as a templating language for very large Markov networks.

Formally, an MLN $\mathcal{M}$ is a knowledge base consisting of a set of FOL formulas $\varphi = \{\varphi_i\}_{i=1}^M$, each with an associated weight $w_i \in \mathbb{R}$. The weights represent the strength of each constraint, where higher weights correspond to increased influence. Thereby, an MLN represents a probability distribution over a finite set of possible worlds $\mathcal{Z}$, where the probability of observing a particular $z \in \mathcal{Z}$ is given by

$$P_{\mathcal{M}}(z) = \frac{1}{Z} \exp\left( \sum_i w_i \varphi_i(z) \right) \quad (2)$$

where, $\varphi_i(z)$ is the number of true groundings of $z$ in $\varphi_i$, and

$$Z = \sum_{z \in \mathcal{Z}} \exp\left(\sum_i w_i \varphi_i(z)\right) \tag{3}$$

is a normalizing constant, also referred to as the *partition function*.

Unlike strict logical rules, MLNs can handle uncertainty or imperfect or contradictory knowledge. This makes them more suitable for settings in which probabilistic associations can more adequately represent the underlying structure of the data.

## 3. OOD Detection with MLNs

As discussed in Section 2.1, most existing OOD detectors focus on identifying atypical patterns within the representations in some layer of a deep neural network trained on ID data. However, certain outliers can be detected more naturally through human-interpretable semantic constraints. For example, using FOL, we can assert that for ID inputs, the following constraint should hold:

$$\forall \mathbf{x} \quad \text{CLASS}(\mathbf{x}) = \texttt{stop\_sign} \rightarrow$$
$$\text{COLOR}(\mathbf{x}) = \texttt{red} \wedge \text{SHAPE}(\mathbf{x}) = \texttt{octagon}, \tag{4}$$

which states that stop signs must appear as red octagons. By describing the properties of the data-generating distribution at this semantic level, the decision-making process becomes more transparent and interpretable to humans.

In our proposed approach, we first use some DNNs to learn the semantic meaning of a set of FOL predicates and functions. Next, we check whether some input violates any weighted logical constraints via a Markov logic network. Such violations change the sample's OOD score, enabling us to detect outliers that deviate from the known semantic structure of ID data while providing a clear rationale for why an input might be OOD. Alg. 1 provides an overview of the score computation, which will be described in the following.

**Connecting DNNs and FOL Semantics**   To apply logical statements like Eq. (4) for OOD detection, we must bridge the gap between the high-level semantics of the used logical predicates and functions and the raw pixel space. In FOL, this connection is achieved through the interpretation $\mathcal{I}$, which determines which predicates evaluate to true for a given input. Because manually specifying these interpretations for high-dimensional inputs is prohibitive, we train DNNs to approximate them. For instance, a neural color-classifier $f_{\text{COLOR}}(\mathbf{x})$ outputs logits for candidate colors, and we take the $\arg\max$ to decide if $\mathbf{x}$ is, e.g., "red". Given a set of DNNs which serve as interpretation $\mathcal{I}$, an input $\mathbf{x}$ can

be represented as a point $z$ in a semantic space $\mathcal{Z}$ such that

$$z = \langle \mathcal{F}_1^{\mathcal{I}}(\mathbf{x}), \ldots, \mathcal{F}_n^{\mathcal{I}}(\mathbf{x}), \mathcal{P}_1^{\mathcal{I}}(\mathbf{x}), \ldots, \mathcal{P}_m^{\mathcal{I}}(\mathbf{x})\rangle, \tag{5}$$

where each dimension corresponds to a human-understandable *concept*. This disentangled representation can then be used to verify constraints imposed on the semantic space. During the computation of the partition function in Eq. (3), we sum over this semantic space $\mathcal{Z}$ instead of the input space of the DNNs.

**Probabilistic Reasoning**   Once the interpretations of all FOL functions and predicates are determined, the truth value of logical formulas can be evaluated to determine whether some input complies with the defined constraints. Enforcing logical constraints strictly, as proposed in (Kirchheim et al., 2024), leads to rejecting all inputs that violate even a single constraint, which is unsuitable for many real-world applications. A more nuanced approach would be to adjust the OOD score based on the extent of constraint violations. For example, constraints that are frequently satisfied within ID data should contribute more heavily to the OOD score when violated.

Markov logic networks provide a powerful framework for such probabilistic reasoning. By defining a probability distribution over the semantic space based on weights learned from ID data, MLNs allow for the integration of semantic reasoning with probabilistic flexibility. This approach alleviates the rigidity of strict logic-based methods while maintaining interpretability and robustness.

### 3.1. Standalone MLNs

Given a set of constraints and corresponding weights, we could directly use the negative probability $-P_{\mathcal{M}}(\mathbf{x})$ as defined by Eq. (2) as outlier score. However, computing the partition function Eq. (3) is computationally demanding, as it involves summation over $\mathcal{Z}$. Since the partition function is identical for all inputs, and strictly monotonic transformations, such as multiplication with a positive factor and exponentiation, will not change the ordering of the outlier scores, we propose to use the following outlier score:

$$D_{\mathcal{M}}(\mathbf{x}) = -\sum_i w_i \varphi_i(\mathbf{x}) . \tag{6}$$

Omitting these operations significantly accelerates inference (see Section 5.4). While this comes at the cost of probabilistic interpretability of the MLN outputs, this is not required for OOD detection. Furthermore, since $D_{\mathcal{M}}$ is just the negative weighted sum of the satisfied constraints, the resulting score can be easily decomposed into the contribution of the individual constraints: if all constraints are satisfied, the outlier score is $-\sum_i w_i$. The violation of constraint $\varphi_i$ will increase the score by $w_i$. Note, however, that weights can

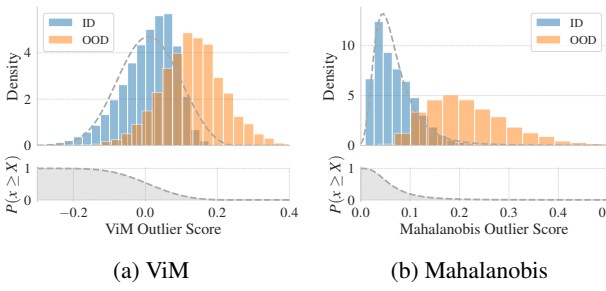

(a) ViM      (b) Mahalanobis

*Figure 1.* Distribution of outlier scores on ID samples from the CelebA dataset and some OOD data. A fitted generalized extreme value distribution (GED) in grey. Survival function below.

be negative, so constraint violations could also effectively decrease the score.

### 3.2. Combining MLNs and existing OOD Detectors

Given a DNN-based interpretation of some predicates, the MLN can determine how plausible the semantic representation of some input is, but it does not directly account for how unusual the input appears in terms of its neural representation. Conversely, common OOD detectors rely on neural representations but neglect explicit semantic constraints. Combining these complementary signals could potentially yield more robust OOD detection.

**Score Normalization**    Different OOD detectors produce outlier scores on different scales, which complicates direct fusion (see Fig. 1). To address this, we propose to normalize an existing detector's scores $D(\mathbf{x})$ into the $[0, 1]$ range by estimating its distribution $p_D$ on ID data. Specifically, for any new input $\mathbf{x}$, we compute the survival function

$$p_D\big(D(\mathbf{x})\big) \;=\; P\big(D(X) \geq D(\mathbf{x})\big), \tag{7}$$

which indicates how extreme the observed score is relative to ID samples.

**Score Combination**    Once the baseline detector's outlier scores are normalized, we multiply them by the MLN-based outlier score, $D_\mathcal{M}(\mathbf{x})$:

$$D'_\mathcal{M}(\mathbf{x}) \;=\; D_\mathcal{M}(\mathbf{x}) \,\times\, p_D\big(D(\mathbf{x})\big). \tag{8}$$

Intuitively, $D_\mathcal{M}(\mathbf{x})$ captures the semantic plausibility of $\mathbf{x}$, while $p_D\big(D(\mathbf{x})\big)$ quantifies how rare $\mathbf{x}$'s neural representation is among ID samples. Since the survival function $p_D$ provides normalized values in a bounded interval, we can omit the costly evaluation of the partition function Eq. (3) for the MLN without affecting the overall ranking of outlier scores and, depending on the choice of distribution family, evaluating $p_D$ is much faster. We note that if $D$ already produces normalized outputs, additional normalization might not be required.

### 3.3. Supervised Training

One of the most effective approaches in OOD detection remains the supervised training of a model with a set of auxiliary outliers (Hendrycks et al., 2018; Dhamija et al., 2018). However, the existing techniques have a disadvantage: They usually reduce the model's classification accuracy on ID data or require special training schemes. They also have the same limitations as outlined in the introductory section.

We can leverage the modularity of our approach by extending an existing MLN with an additional predicate $\mathcal{P}_{\text{ID}}$ for the concept of ID, which can be learned from an auxiliary dataset containing ID and OOD data. Afterwards, we extend the knowledge base with a constraint that asserts that each input should be ID. Compared to existing approaches, this does not require any changes to the remainder of the system.

---

**Algorithm 1** Combined Outlier Score Computation

---

1: **Input:** Input $\mathbf{x}$, constraints $\varphi$ with interpreted functions $\mathcal{F}^\mathcal{I}$ and predicates $\mathcal{P}^\mathcal{I}$, constraint weights $w$, normalization function $p_D$, OOD detector $D$
2: **Output:** Outlier score $o$ for $\mathbf{x}$
3: $o_D \leftarrow D(\mathbf{x})$
4: $o_D^{\text{norm}} \leftarrow p_D(o_D)$
5: $z = \langle \mathcal{F}_1^\mathcal{I}(\mathbf{x}), \dots, \mathcal{F}_n^\mathcal{I}(\mathbf{x}), \mathcal{P}_1^\mathcal{I}(\mathbf{x}), \dots, \mathcal{P}_m^\mathcal{I}(\mathbf{x}) \rangle$
6: $o_\mathcal{M} \leftarrow -\sum_i w_i \varphi_i(z)$
7: $o = o_\mathcal{M} \times o_D^{\text{norm}}$
8: **return** $o$

---

## 4. In-Distribution Constraint Search

Given the absence of prior knowledge about constraints for some domains, we propose an algorithm to automatically learn constraints for OOD detection from data.

A logical constraint, as used here, can be represented as a full binary tree, where the leaves are (possibly negated) predicates or functions with an equality constraint, and the branches are binary logical connectors. An example is provided in Fig. 2. For some sets of predicates and functions, let $\mathcal{T}$ be the set of all possible formulas representable by trees up to a certain depth (possibly subject to additional constraints on the tree's structure). We are then searching for an optimal set of formulas that maximizes the weighted sum of a performance measure $J$ of the resulting detector and an optional regularization term $C$ that penalizes the complexity of a solution candidate to improve the generalization of the found constraint set. Formally, we want to solve

$$\max_{\varphi \in \mathscr{P}(\mathcal{T})} \underbrace{\mathbb{E}_{(\mathbf{x}_{\text{ID}}, \mathbf{x}_{\text{OOD}})} \left[ J(\varphi, \mathbf{x}_{\text{ID}}, \mathbf{x}_{\text{OOD}}) \right]}_{\text{Performance}} - \lambda \underbrace{C(\varphi)}_{\text{Complexity}} \tag{9}$$

where $\mathscr{P}$ denotes the powerset, and $\lambda \in \mathbb{R}_{\geq 0}$ is a balancing weighing factor.

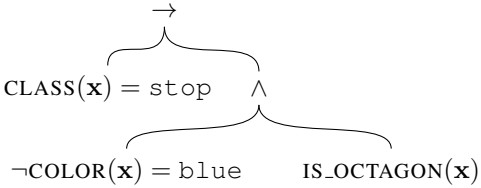

*Figure 2.* Representation of a constraint as a tree of depth 3.

**Performance**   As a measure of performance $J$ of a set of rules $\varphi$, we can use the AUROC of the resulting detector $D$. The AUROC of $D$ can be interpreted as the probability that a randomly drawn ID example receives a higher outlier score than a randomly drawn OOD example:

$$\text{AUROC} \triangleq \mathbb{E}_{(\mathbf{x}_{\text{ID}}, \mathbf{x}_{\text{OOD}})} \left[ \mathbb{1}\{ D(\mathbf{x}_{\text{OOD}}) > D(\mathbf{x}_{\text{ID}}) \} \right] \quad (10)$$

where $\mathbb{1}$ denotes the indicator function.

**Complexity**   As a measure of complexity $C$ of a set of rules $\varphi$, we use the number of constraints $|\varphi|$.

**Optimization**   Directly solving Eq. (9) by exhaustively evaluating all possible subsets of constraints is computationally infeasible for all but the smallest instances, as it would require evaluating $2^{|\mathcal{T}|}$ combinations. To address this, we adopt a greedy search strategy, described in Alg. 2. The procedure begins with an initially empty (or partially specified) set of formulas $\varphi$. At each iteration, a candidate constraint from the remaining pool is considered for inclusion. The augmented set is then used to train a detector on the ID training set $\mathcal{D}_{\text{train}}$ and evaluated on the validation set $\mathcal{D}_{\text{val}}$, which contains both ID and OOD samples to enable empirical estimation of the AUROC. A candidate constraint is retained only if it results in an improvement of the objective value. As we only consider adding a single constraint in each iteration, the complexity penalty implies that we will only accept the current candidate constraint if it improves the AUROC by at least $\delta_{\min} \in \mathbb{R}_{\geq 0}$. In other words, in Alg. 2, $\delta_{\min}$ serves as the $\lambda$ weighting factor of the complexity penalty. While this greedy approach does not guarantee a globally optimal solution, it significantly reduces the computational burden, requiring only $|\mathcal{T}|$ evaluations.

## 5. Experiments

The source code for our experiments is available online.[1]

### 5.1. Implementation & Experimental Setting

**Compiling Constraints**   We implement a compiler that transforms constraints formulated in a human-understandable format, such as `class=stop_sign ->`

---

[1] https://github.com/kkirchheim/mln-ood

---

**Algorithm 2** Greedy Constraint Set Search

1: **Input:** Training set $\mathcal{D}_{\text{train}}$, validation set $\mathcal{D}_{\text{val}}$, baseline performance $J_0$, set of possible constraints $\mathcal{T}$
2: **Output:** Set of constraints $\varphi$
3: Initialize $\varphi \leftarrow \emptyset$
4: Initialize $J \leftarrow J_0$
5: **for** each constraint $\varphi_i \in \mathcal{T}$ **do**
6:     $\varphi' \leftarrow \varphi \cup \{\varphi_i\}$
7:     Train detector with $\varphi'$ on $\mathcal{D}_{\text{train}}$
8:     $J' \leftarrow$ Evaluate detector on $\mathcal{D}_{\text{val}}$
9:     **if** $J' > J + \delta_{\min}$ **then**
10:        $J \leftarrow J'$
11:        $\varphi \leftarrow \varphi'$
12:     **end if**
13: **end for**
14: **return** $\varphi$

---

`color=red and shape=octagon`, into PyTorch operations. These operations can be executed for many inputs in parallel, which allows efficient checking of a large number of points against a constraint during training and inference.

**Choice of Distribution Family**   Empirically, we observe that the distribution of outlier scores of several existing detectors can be modeled with sufficient accuracy by a generalized extreme value distribution (GED) (see Fig. 1), and we will use this distribution family to model the survival function of outlier scores in the following. An ablation study is provided in Section 5.4.

**DNN and MLN Optimization**   During training, we first optimize the parameters of the DNNs by minimizing the cross-entropy on some training data. Subsequently, the MLNs parameters $\mathbf{w}$, which we initialize with $-1$, are optimized by minimizing the negative log-likelihood, as

$$\min_{\mathbf{w} \in \mathcal{W}} \frac{1}{|\mathcal{D}_{\text{ID}}|} \sum_{\mathbf{x} \in \mathcal{D}_{\text{ID}}} -\log P_{\mathcal{M}}(\mathbf{x}) \quad (11)$$

using the L-BFGS optimizer for 10 epochs with a learning rate of 0.01. For large MLNs, optimizing the pseudo-likelihood instead could provide performance speedups (Richardson & Domingos, 2006).

**Out-of-Distribution Data**   As OOD test data, we use images from 8 different sources that cover near and far OOD data, including cropped and resized variants of the LSUN (Yu et al., 2015) and the TinyImageNet datasets, Gaussian and Uniform Noise, Places356 (Zhou et al., 2017), and iNaturalist (Van Horn et al., 2018). During constraint search, we use Textures (Cimpoi et al., 2014) as OOD data. For the supervised variant, we use a database of tiny images (Hendrycks et al., 2018) as OOD training data.

**Seed Replicates** Several works have found that training DNNs several times from different initializations can yield significantly different experimental results (Bouthillier et al., 2019; Summers & Dinneen, 2021). We, therefore, replicate experiments several times with different random seeds to mitigate the effects of statistical fluctuations.

## 5.2. Traffic Sign Classification

The German Traffic Sign Recognition Benchmark (GTSRB) dataset (Stallkamp et al., 2012) contains approximately 40,000 images of German traffic signs spanning 43 classes. While the dataset provides labels for the type of sign, additional labels for the color and shape of each sign are known *a priori*. This prior knowledge can be encoded into a knowledge base containing statements resembling Eq. (4), which associate each traffic sign class with the usual color and shape. Note that no additional labeling is needed.

**Setup** We train different WideResNet-40s (Zagoruyko & Komodakis, 2016) for the functions and predicates of the domain: CLASS, SHAPE, and COLOR. The original training set is split into 35,000 images for training, and 4,209 images for validation, with all pictures resized to $32 \times 32$. The DNNs, which were pre-trained on a downscaled variant of the ImageNet, are then further trained for ten epochs using mini-batch SGD with a Nesterov momentum of 0.9, an initial learning rate of 0.01 with a cosine annealing schedule (Loshchilov & Hutter, 2017), and a batch size of 32. For the supervised variant, we additionally train a DNN for the ID-predicate $\mathcal{P}_{\mathrm{ID}}$ (see Section 3.3).

**Results** The results are listed in Tab. 1. The naive MLN detector Eq. (6) achieves an AUROC $> 86\%$, significantly above random guessing. This demonstrates that many OOD inputs can be detected on a purely semantic level. Furthermore, combining the MLN with other OOD detectors outperforms all purely pattern-based OOD detectors. For example, MLN+Ensemble reduces the FPR95 by $> 37\%$ relative to the ensemble baseline. The observed AUROC improvement for the combined MLN+Ensemble compared to the standalone Ensemble of the individual detectors, while numerically small, is statistically significant (t-test $p < 0.05$) and has a large effect size (Cohen's $d \approx 1.27$). When incorporating auxiliary outliers, the supervised MLN+Ensemble+ achieves almost perfect scores, significantly outperforming all other methods, including the supervised approach based on logical reasoning.

## 5.3. Face Attribute Prediction

The CelebA dataset (Liu et al., 2015) comprises approximately 200,000 images with 40 binary attribute annotations, covering concepts such as gender, age, the presence of facial hair, and more. Compared to the GTSRB dataset, CelebA poses greater challenges for several reasons: Firstly, there is limited prior knowledge about possible constraints governing the combinations of attributes. Secondly, the constraints in this dataset are likely softer than those in the GTSRB. For example, while it is reasonable to assume that individuals with grey hair are not young, there might plausibly be exceptions. Lastly, prior work has identified significant labeling noise in the dataset (Lingenfelter et al., 2022), which likely reduces the effectiveness of strict constraint checking. Consequently, we expect a probabilistic treatment to provide significant advantages over a strictly logical approach.

We apply the proposed Alg. 2 for constraints up to depth 2, constructed from selected 14 predicates. We use the implication ($\rightarrow$) as a logical connector - which, combined with a negation ($\neg$), is functionally complete - and $\delta_{\min} = 0.01$ (*i.e.*, we select a constraint if it increases the validation AUROC by at least 1%). The search over all 702 candidates yielded the following set of constraints:

$$\forall \mathbf{x} \quad \text{YOUNG}(\mathbf{x}) \tag{12}$$

$$\forall \mathbf{x} \quad \text{HEAVY\_MAKEUP}(\mathbf{x}) \rightarrow \text{GRAY\_HAIR}(\mathbf{x}) \tag{13}$$

$$\forall \mathbf{x} \quad \text{WEARING\_LIPSTICK}(\mathbf{x}) \rightarrow \text{GRAY\_HAIR}(\mathbf{x}) \tag{14}$$

$$\forall \mathbf{x} \quad \text{WEARING\_LIPSTICK}(\mathbf{x}) \rightarrow \text{NO\_BEARD}(\mathbf{x}) \tag{15}$$

$$\forall \mathbf{x} \quad \neg\text{MALE}(\mathbf{x}) \rightarrow \text{NO\_BEARD}(\mathbf{x}) \tag{16}$$

**Setup** For this dataset, we use ResNet-18s (He et al., 2016) pre-trained on ImageNet (Russakovsky et al., 2015). All images are resized to $224 \times 224$. We split the data into 150000, 2599, and 50000 images for training, validation, and testing, respectively. Each DNN is trained for ten epochs using mini-batch SGD with a Nesterov momentum of 0.9, an initial learning rate of 0.01 with a cosine annealing schedule (Loshchilov & Hutter, 2017), and a batch size of 32. Again, we additionally train a DNN $f_{\mathrm{ID}}$ to distinguish between ID and OOD data.

**Results** Results for experiments on the CelebA dataset are provided in Tab. 1. MLN, on its own, outperforms random guessing and several other OOD detectors. Combined detectors incorporating an MLN sometimes outperform both the original detector and the MLN. For example, combining the Mahalanobis method and an MLN (MLN+Mahalanobis) reduces the FPR95 by $\approx 20\%$. Again, using auxiliary outliers during training further increases performance.

## 5.4. Ablation Studies & Discussion

**Generalizing to other OOD Detectors** Results for the combination of an MLN with various OOD detectors, including those based on posteriors, logits, latent representations, and rectified latents, are provided in Tab. 3. The results demonstrate that incorporating an MLN-based semantic OOD detection consistently enhances the performance of

*Table 1.* Mean performance and corresponding standard error of different OOD detection methods in our experiments. Results averaged over ten experimental trials with different random seeds. The best result is in bold, and the second best is underlined. ↑ indicates that higher values are better, while ↓ indicates the opposite – all values in percent.

| Detector | Reasoning | Super-vision | AUROC ↑ | | AUPR-ID ↑ | | AUPR-OOD ↑ | | FPR95 ↓ | |
|---|---|---|---|---|---|---|---|---|---|---|
| | | | GTSRB (Stallkamp et al., 2012) | | | | | | | |
| MSP (Hendrycks & Gimpel, 2017) | ✗ | ✗ | 98.96 | ±0.25 | 99.04 | ±0.30 | 96.62 | ±1.19 | 3.04 | ±0.81 |
| EBO (Liu et al., 2020) | ✗ | ✗ | 99.05 | ±0.34 | 98.85 | ±0.39 | 98.28 | ±0.46 | 3.58 | ±1.80 |
| Ensemble (Lakshminarayanan et al., 2017) | ✗ | ✗ | 99.80 | ±0.03 | 99.84 | ±0.05 | 99.35 | ±0.22 | 0.86 | ±0.15 |
| SHE (Zhang et al., 2022) | ✗ | ✗ | 84.13 | ±2.18 | 83.23 | ±4.39 | 84.69 | ±1.63 | 61.00 | ±9.20 |
| Mahalanobis (Lee et al., 2018) | ✗ | ✗ | 99.23 | ±0.08 | 99.55 | ±0.13 | 94.67 | ±2.31 | 1.85 | ±0.27 |
| ViM (Wang et al., 2022) | ✗ | ✗ | 99.47 | ±0.08 | 99.63 | ±0.13 | 97.30 | ±1.18 | 1.71 | ±0.30 |
| DICE (Sun & Li, 2022) | ✗ | ✗ | 99.04 | ±0.35 | 98.83 | ±0.39 | 98.24 | ±0.47 | 3.62 | ±1.82 |
| ReAct (Sun et al., 2021) | ✗ | ✗ | 96.85 | ±0.46 | 98.04 | ±0.53 | 86.54 | ±5.59 | 9.70 | ±1.86 |
| Logic (Kirchheim et al., 2024) | Logical | ✗ | 86.17 | ±0.58 | 91.73 | ±2.24 | 90.72 | ±1.78 | 93.76 | ±4.19 |
| Logic+Ensemble (Kirchheim et al., 2024) | Logical | ✗ | 99.85 | ±0.01 | **99.90** | ±0.03 | 99.35 | ±0.29 | **0.48** | ±0.08 |
| MLN (ours) | Probabilistic | ✗ | 86.16 | ±0.58 | 91.72 | ±2.25 | 89.66 | ±2.47 | 93.76 | ±4.19 |
| MLN+Ensemble (ours) | Probabilistic | ✗ | **99.89** | ±0.02 | **99.90** | ±0.03 | **99.57** | ±0.17 | 0.54 | ±0.10 |
| MLN+Mahalanobis (ours) | Probabilistic | ✗ | 99.71 | ±0.03 | 99.79 | ±0.07 | 98.50 | ±0.70 | 1.20 | ±0.17 |
| Logic+Ensemble+ (Kirchheim et al., 2024) | Logical | ✓ | 99.82 | ±0.00 | 99.93 | ±0.01 | 99.32 | ±0.32 | 0.36 | ±0.00 |
| MLN+Ensemble+ (ours) | Probabilistic | ✓ | **99.95** | ±0.00 | **99.97** | ±0.01 | **99.74** | ±0.12 | **0.13** | ±0.01 |
| MLN+Mahalanobis+ (ours) | Probabilistic | ✓ | 99.93 | ±0.01 | 99.95 | ±0.02 | **99.74** | ±0.11 | 0.26 | ±0.06 |
| | | | CelebA (Liu et al., 2015) | | | | | | | |
| MSP (Hendrycks & Gimpel, 2017) | ✗ | ✗ | 48.68 | ±4.09 | 85.06 | ±3.29 | 13.42 | ±2.89 | 85.59 | ±9.54 |
| EBO (Liu et al., 2020) | ✗ | ✗ | 45.24 | ±4.60 | 82.56 | ±5.62 | 12.96 | ±3.03 | 77.22 | ±7.69 |
| Ensemble (Lakshminarayanan et al., 2017) | ✗ | ✗ | 83.43 | ±2.35 | 95.21 | ±1.86 | 48.51 | ±4.02 | 43.09 | ±7.23 |
| SHE (Zhang et al., 2022) | ✗ | ✗ | 39.78 | ±2.91 | 82.63 | ±5.38 | 12.22 | ±3.05 | 77.70 | ±5.40 |
| Mahalanobis (Lee et al., 2018) | ✗ | ✗ | 95.12 | ±1.18 | 98.72 | ±0.36 | 80.68 | ±3.78 | 17.77 | ±3.98 |
| ViM (Wang et al., 2022) | ✗ | ✗ | 84.94 | ±3.82 | 93.75 | ±2.93 | 69.62 | ±6.51 | 48.59 | ±11.30 |
| DICE (Sun & Li, 2022) | ✗ | ✗ | 46.83 | ±4.85 | 83.00 | ±5.59 | 13.53 | ±2.97 | 76.05 | ±7.92 |
| ReAct (Sun et al., 2021) | ✗ | ✗ | 44.84 | ±4.72 | 82.29 | ±5.50 | 12.69 | ±3.03 | 77.53 | ±8.37 |
| Logic (Kirchheim et al., 2024) | Logical | ✗ | 51.06 | ±3.59 | 85.86 | ±3.98 | 44.24 | ±3.05 | 91.64 | ±5.48 |
| Logic+Ensemble (Kirchheim et al., 2024) | Logical | ✗ | 54.46 | ±2.92 | 86.84 | ±3.87 | 44.96 | ±2.98 | 76.55 | ±2.75 |
| MLN (ours) | Probabilistic | ✗ | 84.13 | ±2.40 | 96.31 | ±1.14 | 57.88 | ±4.40 | 41.63 | ±6.71 |
| MLN+Ensemble (ours) | Probabilistic | ✗ | 90.42 | ±1.73 | 97.48 | ±0.89 | 65.23 | ±4.12 | 24.84 | ±4.77 |
| MLN+Mahalanobis (ours) | Probabilistic | ✗ | **96.01** | ±0.94 | **99.01** | ±0.28 | **80.98** | ±3.01 | **14.17** | ±3.18 |
| Logic+Ensemble+ (Kirchheim et al., 2024) | Logical | ✓ | 64.30 | ±2.25 | 91.38 | ±4.01 | 57.43 | ±1.88 | 68.73 | ±2.67 |
| MLN+Ensemble+ (ours) | Probabilistic | ✓ | 97.42 | ±1.72 | 98.81 | ±0.98 | **94.23** | ±3.52 | 8.34 | ±5.23 |
| MLN+Mahalanobis+ (ours) | Probabilistic | ✓ | **97.86** | ±0.42 | **99.41** | ±0.25 | 87.41 | ±1.32 | **7.82** | ±1.75 |

OOD detectors based on neural representations alone. This suggests that probabilistic reasoning over semantics offers a complementary signal for OOD detection, distinct from the pattern-based signals identified by these detectors. Furthermore, supervised training yields additional improvements, confirming the effectiveness of our approach in increasing detection performance.

**Generalization to other DNNs**  To evaluate the MLN's generalizability across different backbones, we conducted experiments using several DNN-based feature encoders in an otherwise identical experimental setting. The results, presented in Tab. 2, show the performance on the GTSRB dataset. As anticipated, more powerful backbones yield superior performance for the ensemble baseline as well as the standalone MLN. Integrating MLN with the ensemble method consistently outperforms both individual approaches.

**Parameter Sharing**  Sharing parameters between DNNs decreases computational and memory requirements but could lead to correlated prediction errors, which would intuitively be detrimental to the effectiveness of our approach. To test this hypothesis, we train DNNs on the GTSRB dataset using a linear classifier $h_i$ on top of a shared WideResNet-40 encoder $\Phi$ such that the DNNs $f_i = h_i \circ \Phi$ that constitute the interpretations of the used logical predicates and functions share parameters. Performance measurements averaged over several training runs are listed in Tab. 4. We observe a significant drop in the performance of MLN- and Logic-based methods compared to models without shared parameters, as expected. Still, the MLN-based approach outperforms logical constraint checking.

**Omitting Constraints**  We would expect detection performance to increase as constraints are incrementally added to the system. Results for the GTSRB with varying numbers of constraints are shown in Fig. 3. As we can see, the MLN with a small number of rules barely outperforms random guessing. However, as the number of rules increases, we observe a monotonic improvement in performance. The results further indicate that certain constraints have a more significant impact on the MLN's performance than others. Here, the rule for the stop sign particularly increases performance.

*Table 2.* Performance of methods with different encoders on GTSRB. Results averaged over ten seed replicates. For the Vision Transformer, images have been resized to $224 \times 224$.

| Model | ResNet-18 (He et al., 2016) | | ConvNext Tiny (Liu et al., 2022) | | WideResNet-40 (Zagoruyko & Komodakis, 2016) | | ViT (Dosovitskiy et al., 2021) | |
|---|---|---|---|---|---|---|---|---|
| | AUROC ↑ | FPR95 ↓ | AUROC↑ | FPR95 ↓ | AUROC ↑ | FPR95 ↓ | AUROC ↑ | FPR95 ↓ |
| Ensemble | 96.8 | 10.1 | 99.1 | 3.9 | 99.8 | 0.85 | **99.9** | 0.28 |
| MLN (ours) | 81.9 | 97.5 | 84.4 | 100.0 | 86.2 | 93.8 | 90.3 | 82.5 |
| MLN+Ensemble (ours) | **98.0** | **8.3** | **99.5** | **2.9** | **99.9** | **0.54** | **99.9** | **0.17** |

*Table 3.* AUROC for different detectors using a patter-based baseline detector, combination with MLN, and a supervised MLN-based detector. All values in percent, results averaged over ten different random seeds. $\Delta$ indicates the difference to the next left column.

| Detector | Detector | +MLN | +Supervision |
|---|---|---|---|
| | GTSRB (Stallkamp et al., 2012) | | |
| MSP | 98.96 | 99.60 △ 0.64 | 99.90 △ 0.30 |
| Ensemble | 99.80 | 99.88 △ 0.08 | 99.96 △ 0.08 |
| EBO | 99.05 | 99.50 △ 0.45 | 99.77 △ 0.27 |
| DICE | 99.04 | 99.50 △ 0.46 | 99.77 △ 0.27 |
| SHE | 84.13 | 95.04 △ 10.91 | 99.83 △ 4.79 |
| ReAct | 96.85 | 99.09 △ 2.24 | 99.92 △ 0.82 |
| Mahalanobis | 99.23 | 99.72 △ 0.49 | 99.96 △ 0.23 |
| ViM | 99.47 | 99.80 △ 0.33 | 99.96 △ 0.16 |
| | CelebA (Liu et al., 2015) | | |
| MSP | 48.68 | 60.72 △ 12.04 | 71.10 △ 10.38 |
| Ensemble | 83.43 | 90.42 △ 6.99 | 97.42 △ 7.00 |
| EBO | 45.24 | 73.89 △ 28.65 | 89.89 △ 16.00 |
| DICE | 46.83 | 74.98 △ 28.16 | 90.31 △ 15.32 |
| SHE | 39.78 | 71.54 △ 31.76 | 89.75 △ 18.21 |
| ReAct | 44.84 | 72.06 △ 27.22 | 89.55 △ 17.49 |
| Mahalanobis | 95.12 | 96.01 △ 0.89 | 97.86 △ 1.85 |
| ViM | 84.94 | 91.75 △ 6.82 | 97.12 △ 5.37 |

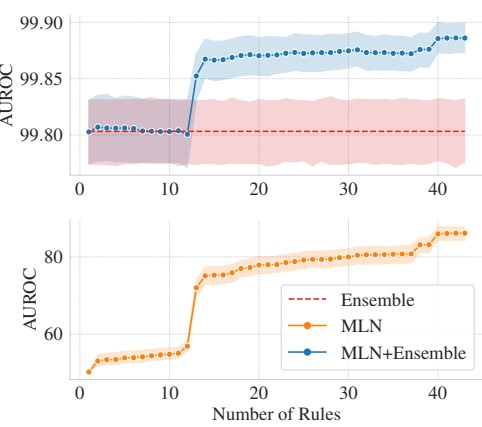

*Figure 3.* OOD detection performance on GTSRB for different numbers of constraints.

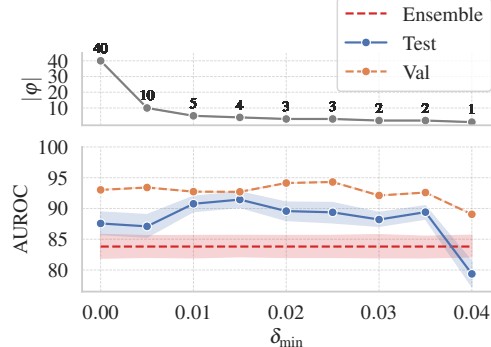

*Figure 4.* Number of identified constraints and corresponding MLN+Ensemble performance for greedy constraint search on CelebA with varying regularization weight $\delta_{\min}$.

**Constraint Search Regularization** By varying the weight of the regularizer $\delta_{\min}$, we can trade off the complexity of the found rule set and its performance. An example is provided in Fig. 4: for $\delta_{\min} = 0$, that is, without regularization, the algorithm identifies 40 constraints but overfits. Increased regularization decreases the number of rules and reduces overfitting up to a certain limit, after which the performance deteriorates again.

**Computational Overhead** Fig. 5 depicts inference time and batch size on the GTSRB, averaged over 100 batches on an Nvidia A100. Checking the 43 constraints introduces a moderate computational overhead of several milliseconds per batch. Omitting the computation of the partition function reduces the overhead. For batch sizes $> 128$, the inference time is dominated by the computation required by DNNs because constraint checking is memory efficient and can be parallelized, leading to a constant overhead, even for large batch sizes.

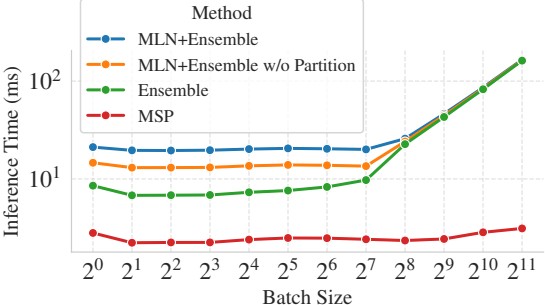

*Figure 5.* Inference time and batch size for GTSRB with 43 constraints. The MLN introduces a moderate computational overhead during inference compared to the ensemble.

Table 4. Performance on GTSRB for models sharing parameters between DNNs. The Δs indicate the difference to models without parameter sharing. All values in percent.

| Detector | AUROC ↑ | | FPR95 ↓ | |
|---|---|---|---|---|
| Ensemble | 99.14 | Δ -0.67 | 3.28 | Δ 2.42 |
| Logic | 47.69 | Δ -38.47 | 100.00 | Δ 6.24 |
| Logic+Ensemble | 56.96 | Δ -42.90 | 60.01 | Δ 59.53 |
| MLN | 53.61 | Δ -32.55 | 100.00 | Δ 6.24 |
| MLN+Ensemble | 98.88 | Δ -1.01 | 3.67 | Δ 3.19 |

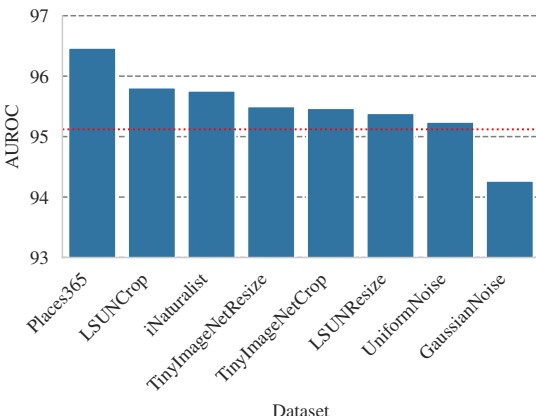

Figure 6. Influence of OOD dataset used for constraint search on evaluation performance for MLN+Mahalanobis. Mahalanobis baseline performance is marked in red.

**Influence of OOD Dataset** The constraints discovered by the search algorithm are sensitive to the specific OOD dataset used during optimization. To investigate this dependency, we performed a constraint search independently for each OOD dataset included in our evaluation, and subsequently assessed the performance of the resulting MLN+Mahalanobis detector on CelebA. Importantly, during evaluation, the OOD dataset used for the constraint search was excluded to simulate realistic generalization to unseen outliers. The results, shown in Fig. 6, indicate that performance typically surpasses the baseline, with the notable exception of Gaussian Noise. This suggests that the diversity and realism of the training-time OOD samples play a role in learning effective constraints.

**Influence of Normalization** Tab. 5 presents the effect of different distribution families used to normalize outlier scores on detection performance. Among the tested options, the GED yields the highest AUROC scores across both MLN+Ensemble and MLN+ViM detectors. In contrast, omitting normalization leads to a substantial performance drop. These results highlight the critical role of selecting a well-matched distributional prior when modeling outlier scores.

Table 5. Effect of the distribution family used for outlier score normalization on AUROC (in percent). Higher values indicate better performance.

| Distribution | MLN+Ensemble | MLN+ViM |
|---|---|---|
| GED | 99.88 | 99.80 |
| Uniform | 99.76 | 99.46 |
| Normal | 99.07 | 98.13 |
| Generalized Normal | 98.67 | 99.63 |
| LogNormal | 98.60 | 99.79 |
| No Normalization | 86.16 | 86.16 |

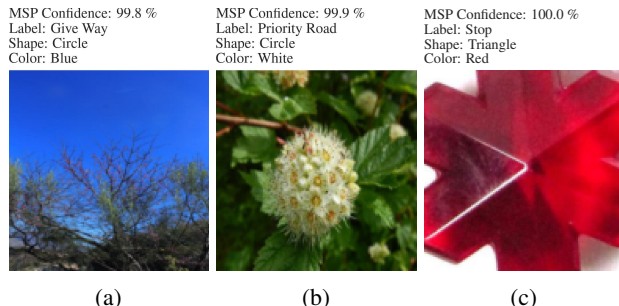

(a)  (b)  (c)

Figure 7. OOD samples with MSP confidence as predicted by a DNN trained on the GTSRB dataset.

**Explainability** Fig. 7 depicts a sample of OOD images for the GTSRB, together with the detected concepts and confidence predicted by the MSP baseline. Since the detected combination of concepts violates the constraint for the corresponding traffic sign, our method would increase the outlier score, and this increase can be traced back to specific rules. For example, in Fig. 7c, the constraint that a stop sign implies an octagonal shape is violated. Since this constraint's weight is ≈ 4.89, the violation of this constraint increases the outlier score the MLN assigns to the input by the same amount.

## 6. Conclusion

In this work, we presented a novel framework for OOD detection based on Markov logic networks that probabilistically verifies logical constraints over a low-dimensional semantic representation of a given input. This approach can be combined with various existing OOD detectors and backbones to increase OOD detection performance while providing a certain degree of interpretability. We also introduced a simple algorithm that can be used to automatically derive such logical constraints from a dataset.

## Acknowledgment

This research received funding from the *Federal Ministry for Economic Affairs and Climate Action (BMWK)* and the *European Union* under grant agreement 19I21039A.

## Impact Statement

Advances in OOD detection, particularly in Neuro-Symbolic approaches, could potentially benefit the deployment of safe Machine Learning-driven systems in the real world. While the societal impact of this is broad, we do not consider it specific to our work.

The proposed algorithm Alg. 2 for learning constraints for OOD detection introduces the possibility of embedding discriminatory biases against underrepresented minority classes into the broader system. For instance, in the Face Attribute Prediction use case analyzed in our experiments, the learned in-distribution constraints, such as

$$\forall \mathbf{x} \quad \neg \text{MALE}(\mathbf{x}) \rightarrow \text{NO\_BEARD}(\mathbf{x})$$

may be considered culturally or contextually insensitive, as they might fail to capture the diversity of the population outside of the training distribution.

While this is a general problem in machine learning, compared to prior works, Markov logic networks offer several advantages in addressing these sensitive scenarios:

- Since the rules governing the detector's behavior are explicit and human-understandable (as opposed to encoded in the weights of a connectionist system), biases are more apparent and can be more easily identified and addressed.

- Compared to methods like LogicOOD (Kirchheim et al., 2024), the employed probabilistic approach is able to learn weights flexibly, which allows the algorithm to account for violations of constraints given adequate representation of such cases in the training dataset.

- Furthermore, manually adjusting individual constraint weights is possible and has an intuitive interpretation.

Therefore, we believe that the transparency of the presented approach advances its potential to comply with ethical standards.

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
