# OpenReview forum: "Improving Out-of-Distribution Detection with Markov Logic Networks"
_ICML.cc/2025/Conference — ICML 2025 poster_

### Official Review · Reviewer_oa1P · 2025-03-11

**Overall Recommendation:** 2

**Summary:**

This paper proposes a novel framework for improving out-of-distribution (OOD) detection using Markov Logic Networks (MLNs), which combine probabilistic reasoning with human-interpretable logical constraints. The approach addresses limitations of traditional OOD detectors, such as reliance on superficial statistical patterns and lack of explainability. Empirical validation across benchmarks, demonstrating superior performance and computational efficiency.

**Claims And Evidence:**

Yes.

**Essential References Not Discussed:**

No.

**Experimental Designs Or Analyses:**

Yes.

**Methods And Evaluation Criteria:**

Yes.

**Other Comments Or Suggestions:**

Please refer to the weaknesses above.

**Other Strengths And Weaknesses:**

Strengths:
1. The authors propose using Markov Logic Networks (MLNs) to combine first-order logic constraints with probabilistic reasoning. This allows the model to incorporate human-understandable semantic constraints (e.g., "stop signs are red and octagonal") into the detection process, improving explainability and enabling the integration of prior knowledge.
2. The framework combines MLN-derived semantic scores with traditional neural representation-based detectors through score normalization and multiplication. This hybrid approach leverages both semantic plausibility and neural pattern recognition, significantly boosting performance. Also, the authors propose an algorithm to automatically learn logical constraints from data when prior knowledge is unavailable. The greedy search strategy selects constraints that maximize detection performance while regularizing complexity, ensuring robustness against overfitting.
3. Extensive experiments on datasets like GTSRB (traffic signs) and Celeb-A (face attributes) demonstrate that MLN-augmented detectors achieve higher AUROC and lower FPR95 scores compared to standalone methods.

Weaknesses:
1. The inline formula in Line 181 is missing a label and fails to clearly explain how the specific values are calculated.
2. Section 4 needs to be reorganized and polished, as some sections lack clarity. For example, the phrase "maximizes the weighted sum of the discriminative power J" leaves readers unclear about the meaning of "discriminative power."
3. There is a lack of explanation and transition for the interpretation of C(\phi) in Eq. (8). For example, why we need to penalize the complexity of a solution candidate, how is it calculated, and what is its relationship to Eq. (9)?
4. The authors proposes a greedy search strategy outlined in Alg. 1 to solving Eq. (8). However, why the strategy could solve the optimization of Eq. (8) is unclear.
5. In Section 1, it mentions that “neuro-symbolic approaches for OOD detection have shown promise in addressing these limitations”. This suggests that the work represents an incremental advance over prior efforts by employing a different model to express logical rules in the context of OOD detection.
6. Additionally, it is recommended to provide a complete algorithmic workflow rather than focusing solely on the algorithmic steps of a minor module. Alternatively, present an overall algorithmic framework first, and if the detailed steps of a specific component are complex, include supplementary algorithmic details for clarity.

**Questions For Authors:**

1. In Line 274, it mentions that “Additionally, we train a DNN for the ID-predicate”. The training methodology for this component is not described, and it remains unclear whether it follows the same training paradigm as the previously mentioned network.

**Relation To Broader Scientific Literature:**

The key contributions of the paper are closely related to the fields of out-of-distribution (OOD) detection and Markov Logic Networks (MLNs). The authors propose using MLNs to combine first-order logic constraints with probabilistic reasoning, allowing the model to incorporate human-understandable semantic constraints into the OOD detection process.

**Theoretical Claims:**

Not applicable, as this paper does not put forward any theoretical claims or provide corresponding proofs.

---

> ### Author Rebuttal · Authors · 2025-03-25
>
> We sincerely thank the Reviewer for the comprehensive and constructive feedback. Below, we address the specific points raised by the reviewer.
>
> ### Comment 2
>
> > the phrase "maximizes the weighted sum of the discriminative power J" leaves readers unclear about the meaning of "discriminative power."
>
> The term "discriminative power" is an established concept in machine learning and refers to the classifier's ability to distinguish between classes - in our case, ID and OOD. For binary classification, common performance measures include accuracy and AUROC, with AUROC being particularly prevalent for OOD detection tasks. This definition was explicitly stated immediately following the cited phrase:
>
> > "As a measure of the discriminative power of the resulting OOD detector , we use the AUROC of the trained detector."
>
> A formal description of AUROC follows this explanation.
>
> ### Comment 3
>
> >  why we need to penalize the complexity of a solution candidate, how is it calculated, and what is its relationship to Eq. (9)?
>
> The complexity penalty serves as regularization to avoid overfitting by limiting model complexity. As described in Algorithm 1, we achieve this by only adding rules that enhance performance by at least a threshold $\delta_{min}$. Appendix A formally demonstrates that this regularization corresponds to using the number of rules $| \varphi |$ as a measure of complexity. We agree with the reviewer that this relationship could be clarified further and will explicitly include this clarification in a revised manuscript.
>
> Figure 4 in our ablation study illustrates that omitting the complexity penalty leads to models with high complexity (many rules) and decreased generalization performance.
>
> Eq. (9) is the definition of the AUROC, which is the measure of discriminative power, $J$, that we optimize for.
>
> ### Comment 4
>
> > The authors proposes a greedy search strategy outlined in Alg. 1 to solving Eq. (8). However, why the strategy could solve the optimization of Eq. (8) is unclear.
>
> Indeed, solving Equation (8) exactly is computationally intractable, as we would have to evaluate $2^{702}$ solutions. Our proposed greedy algorithm runs for $\approx 1$ hour (702 evaluations), and empirical results demonstrate that it finds solutions yielding strong performance. However, we concur (and do not claim otherwise) that this approach does not guarantee a global optimum compared to exhaustive search strategies, which would be computationally prohibitive.
>
> ### Comment 5
>
> > This suggests that the work represents an incremental advance over prior efforts by employing a different model to express logical rules in the context of OOD detection.
>
> We respectfully disagree with this characterization. Previous research employed strictly logical rules, whereas our method introduces probabilistic rules through Markov Logic Networks, providing significantly greater expressive power. The advantage of probabilistic over purely logical approaches is clearly demonstrated by our results on Celeb-A, where LogicOOD (Kirchheim et al.) fails, performing no better than random chance, while our method achieves state-of-the-art results.
>
> Moreover, the recent survey "Recent Advances in OOD Detection: Problems and Approaches" references only LogicOOD (Kirchheim et al.) within this methodological paradigm. We explicitly compare against this baseline and surpass it in all experimental setups, underscoring the novel and substantial contribution of our work.
>
> ### Comment 6
>
> > present an overall algorithmic framework first, and if the detailed steps of a specific component are complex, include supplementary algorithmic details for clarity.
>
> We kindly request the reviewer to consider that we have provided complete source code for all experiments as supplementary material, ensuring transparency and reproducibility. Due to space constraints, explicitly detailing every algorithmic aspect within the main paper has been challenging.
>
> To clarify, our method follows a straightforward process:
>
> 1. We first pass the input x through several deep neural networks (DNNs).
> 2. Based on the outputs of these DNNs, we calculate a baseline outlier score.
> 3. We normalize this baseline outlier score by estimating the probability that ID inputs receive an outlier score that is greater $P(D(x) > X)$.
> 4. Concurrently, we compute a separate outlier score using weights of our trained MLN.
> 5. Finally, we obtain our overall outlier score by multiplying these two scores.
>
> In the additional page permitted for revisions, we will include an explicit and concise algorithmic description of these steps to further enhance clarity.
>
> ### Question 1
> > [...] “Additionally, we train a DNN for the ID-predicate”. The training methodology for this component is not described, [...]
>
> The ID-predicate is produced by a DNN component trained in a supervised manner, as described in Section 3.3.
> We agree that this connection was not sufficiently explicit and will clearly state this in the revision.

---

### Official Review · Reviewer_8kcf · 2025-03-13

**Overall Recommendation:** 3

**Summary:**

The paper presents a novel approach to OOD detection by integrating Markov Logic Networks (MLNs) with existing OOD detectors. This fusion of probabilistic reasoning with logical constraints over human-understandable concepts distinguishes it from traditional statistical or neural representation-based methods.

**Claims And Evidence:**

+ The authors suggest that MLNs provide "improved explainability" due to their use of human-understandable constraints. Yet, this is poorly substantiated. Figure 6 offers an anecdotal example of an OOD decision, but there’s no systematic demonstration or metric.
+ The claim of maintaining efficiency is contradicted by Figure 5, where inference time for a batch size of 1 rises from ~2ms (baseline) to ~10ms (MLN-augmented), a 5x increase. While the overhead diminishes with larger batches, this undermines the efficiency claim for real-time or resource-constrained applications
+ The assertion of "significant" performance improvement is overstated. Table 1 shows that for the GTSRB dataset, the AUROC increases from 99.8% (Ensemble) to 99.9% (MLN+Ensemble), a marginal 0.1% gain.

**Essential References Not Discussed:**

The paper cites neuro-symbolic works (e.g., Besold et al., 2021) but engages superficially, missing a detailed discussion of how MLNs differ from or build on prior probabilistic or symbolic OOD approaches.

**Experimental Designs Or Analyses:**

+ The supervised variant uses auxiliary outliers, but the choice of these outliers (e.g., tiny images) is not justified, nor is their impact on performance explored.
+ For Celeb-A, the constraint search uses a validation set with OOD data (Textures), potentially overfitting the constraints to this specific OOD type. The paper does not test against other OOD distributions (e.g., iNaturalist, LSUN) during constraint learning, risking poor generalization.

**Methods And Evaluation Criteria:**

+ Testing is confined to two datasets, which is insufficient to claim broad applicability. GTSRB (traffic signs) and Celeb-A (faces) are visually distinct and relatively simple compared to diverse OOD challenges (e.g., ImageNet). This narrow scope questions the method’s robustness.
+ Algorithm 1 employs a greedy strategy to select constraints, but its optimality is not justified against alternatives (e.g., exhaustive search, reinforcement learning). The paper acknowledges this limitation superficially without exploring its impact, weakening confidence in the method’s effectiveness.

**Other Comments Or Suggestions:**

+ Beyond "cutlier" and "Mathemetically," minor grammatical errors persist (e.g., "then ID data points" in Section 2.1 should be "than"). These suggest a lack of polish.
+ An ablation on components (e.g., normalization, GED choice) would clarify their contributions.

**Other Strengths And Weaknesses:**

**Strengths**:
+ The concept of blending semantic reasoning with OOD detection is creative and could inspire future work.
+ The constraint search algorithm (Algorithm 1) offers a practical step toward automating constraint derivation, a potential asset in knowledge-scarce domains.

**Questions For Authors:**

Refer to the above comments.

**Relation To Broader Scientific Literature:**

The paper positions itself as an advance over statistical OOD detectors and neuro-symbolic methods like LogicOOD (Kirchheim et al., 2024) by introducing probabilistic MLNs.

**Theoretical Claims:**

The paper does not introduce new theoretical proofs or claims, relying instead on established MLN theory applied to OOD detection.

---

> ### Author Rebuttal · Authors · 2025-03-26
>
> ### Explainability
> Quantitatively measuring explainability remains notoriously challenging, thus we rely primarily on qualitative and structural justification. However, we would highly appreciate if the reviewer could suggest some metrics.
>
> By construction, our method's explainability stems directly from Equation 6, which explicitly represents the outlier score as a weighted sum of human-understandable constraints. This structure allows clear statements regarding by what amount each constraint violation or satisfaction increased or decreased the outlier score.
>
> Additionally, we exemplify the practical benefits of our explainable rule sets explicitly in the "Impact" section.
>
> ### Significance of Improvement
>
> > The assertion of "significant" performance improvement is overstated. Table 1 shows that for the GTSRB dataset, the AUROC increases from 99.8% (Ensemble) to 99.9% (MLN+Ensemble), a marginal 0.1% gain.
>
> We respectfully disagree. The improvement is significant for two reasons:
>
> 1. The improvement is statistically significant (t-test $p < 0.05$) with a Cohen's D of $\approx 1.27$, which indicates an effect size that is considered "very large" by conventional standards.
> 2. At a baseline performance (99.8% AUROC), a 0.1% absolute increase means effectively halving the remaining error rate.
>
> ### Latency
>
> > The claim of maintaining efficiency is contradicted by Figure 5, where inference time for a batch size of 1 rises from ~2ms (baseline) to ~10ms (MLN-augmented), a 5x increase. [...] this undermines the efficiency claim for real-time or resource-constrained applications
>
> The reviewer compares our method with the simplest baseline (MSP). However, this comparison neglects substantial accuracy improvements: FPR95 error decreases from approximately 3% (MSP) to 0.5% (MLN) - a 83 % error reduction.
>
> As Figure 5 shows, the primary overhead originates from using an ensemble rather than the MLN itself. Compared to the Ensemble, our method reduces FPR95 from 0.86% to 0.54% (a 37% reduction) at a cost of 2ms (+20%) per image.
>
> We kindly emphasize that even under the slowest inference scenario (batch size 1), our method achieves more than 80 frames per second, arguably sufficient for typical real-time application requirements.
>
> ### Ablation on OOD Training Distribution
> Our experiments confirm that constraints extracted from datasets exhibiting sufficient complexity (e.g., Textures, iNaturalist, ...) generalize reliably across various OOD scenarios, while constraints mined from e.g., Gaussian Noise, fail to generalize.
>
> We conducted additional experiments to systematically assess this concern: constraints were mined using specific OOD datasets, explicitly excluding those same datasets from subsequent evaluation.
>
> | Dataset        | Method          | AUROC | FPR95 |
> | -------------- | --------------- | ----- | ----- |
> | iNaturalist    | Ensemble        | 84.06 | 42.54 |
> | iNaturalist    | MLN+Ensemble    | 87.44 | 33.52 |
> | iNaturalist    | Mahalanobis     | 95.20 | 17.66 |
> | iNaturalist    | MLN+Mahalanobis | 95.21 | 17.07 |
> | Places365      | Ensemble        | 84.76 | 41.17 |
> | Places365      | MLN+Ensemble    | 86.14 | 27.90 |
> | Places365      | Mahalanobis     | 95.34 | 17.07 |
> | Places365      | MLN+Mahalanobis | 96.06 | 13.89 |
> | Gaussian Noise | Ensemble        | 82.70 | 47.46 |
> | Gaussian Noise | MLN+Ensemble    | 79.74 | 60.19 |
> | Gaussian Noise | Mahalanobis     | 94.81 | 19.51 |
> | Gaussian Noise | MLN+Mahalanobis | 94.99 | 18.91 |
>
> ### Alternative Constraint Optimisers
> We discuss that exhaustive search is infeasible for realistic problems due to computational constraints. While we concur that more sophisticated search strategies might yield improved results, our straightforward greedy approach already achieves state-of-the-art performance and generalises effectively across multiple DNNs and OOD datasets.
>
> ### Auxiliary Outlier Distribution
> We selected the "tiny images" dataset as auxiliary outliers because it is well-established and commonly used in relevant literature, including foundational works such as Outlier Exposure by Hendrycks et al., Energy-based OOD Detection by Liu et al., and the related work on LogicOOD by Kirchheim et al. Thus, the precedent for using this dataset is firmly established.
>
> ### Ablation on Score Normalisation
> Here are results (averaged over 10 random seeds) on GTSRB with different distributions choices for Mahalanobis and ViM:
>
> | Normalization      | MLN+Mahalanobis | MLN+ViM |
> | ------------------ | --------------- | ------- |
> | No Normalization   | 72.57           | 35.25   |
> | Normal Dist.       | 96.89           | 98.13   |
> | Uniform Dist.      | 99.21           | 99.46   |
> | Log Normal         | 99.67           | 99.79   |
> | Generalized Normal | 99.08           | 99.63   |
> | GED                | 99.72           | 99.80   |

---

> > ### Comment · Reviewer_8kcf · 2025-04-03
> >
> > Thank you for the detailed rebuttal and the additional results. The results are compelling. I will raise my score.

---

### Official Review · Reviewer_cnXo · 2025-03-13

**Overall Recommendation:** 3

**Summary:**

This work explores enhancing out-of-distribution (OOD) detection using Markov Logic Networks (MLNs), which integrate probabilistic reasoning with logical constraints for improved structure and interpretability. The proposed framework augments existing OOD detectors by incorporating MLNs to define human-understandable logical constraints, enhancing both detection accuracy and explainability. Additionally, a greedy algorithm is introduced to automatically learn logical constraints from data.

**Claims And Evidence:**

Yes.

**Essential References Not Discussed:**

The work discusses relevant works.

**Experimental Designs Or Analyses:**

Yes.

**Methods And Evaluation Criteria:**

Yes.

**Other Comments Or Suggestions:**

Here are some typos:

line 110: "refered to" -> "referred to"
line 96: "were higher weights correspond" -> "where higher weights correspond"

**Other Strengths And Weaknesses:**

Strengths:

1. Integrating MLNs with OOD detection is a novel approach.

2. Traditional OOD detection methods (e.g., MSP, Mahalanobis distance) provide only a confidence score.
This paper incorporates logical constraints (e.g., "A stop sign should be red") and enhances them with MLNs, improving the explainability of the detection process.

Weaknesses:

1. MLNs introduce additional computational overhead to existing deep learning models. Evaluating constraints and computing MLN weights increase inference time, as shown in Fig. 5 (Inference Time vs. Batch Size). While the paper omits the partition function (Z) in Equation (3) to accelerate inference, this compromises probabilistic interpretability.

2. The evaluation is limited to two datasets: GTSRB (German Traffic Sign Recognition Benchmark) and Celeb-A (Face Attribute Prediction), both featuring clear, discrete attributes (e.g., "STOP sign is RED," "Male vs. Female"). However, the study does not test other OOD detection benchmarks such as CIFAR-10 vs. SVHN or ImageNet-O. Additionally, no NLP or tabular datasets are considered.

**Questions For Authors:**

1. How does the algorithm handle contradictory constraints?

2. Have you tested this method on natural images, NLP or tabular data?

**Relation To Broader Scientific Literature:**

This work builds upon and extends previous research by enhancing OOD detection, a crucial aspect of ensuring the reliability of deep learning models in real-world applications. By referencing and innovating upon existing OOD detection techniques, the paper introduces a novel approach that integrates probabilistic reasoning with logical constraints.

**Theoretical Claims:**

The paper does not have theoretical contribution.

---

> ### Author Rebuttal · Authors · 2025-03-26
>
> We sincerely thank the Reviewer for the comprehensive and constructive feedback. Below, we address the specific points raised by the reviewer.
>
> ### Contradictory Constraints
> > How does the algorithm handle contradictory constraints?
>
> During training, the algorithm can automatically down-weights the conflicting constraints. As a result, the network balances and resolves contradictions based on their relative statistical support in the data.
>
> ### NLP or Tabular
> > Have you tested this method on natural images, NLP or tabular data?
>
> We seek clarification regarding the term "natural images," as we have conducted experiments on image datasets.
>
> Regarding NLP and tabular data, we have not yet tested our approach in these domains. We would greatly appreciate any recommendations from the reviewer regarding specific NLP or tabular datasets they consider valuable for additional evaluation.
>
> ### Other Image Datasets
> >  However, the study does not test other OOD detection benchmarks such as CIFAR-10 vs. SVHN or ImageNet-O.
>
> Indeed, our approach relies explicitly on underlying "structure" or "prior knowledge" inherent to the training data. Such structure is, to our knowledge, absent in benchmarks like CIFAR-10 vs. SVHN or ImageNet-O, as these datasets do not provide semantic constraints or structured knowledge our MLN-based approach is designed to leverage.
>
> ### Latency
> > Evaluating constraints and computing MLN weights increase inference time, as shown in Fig. 5 (Inference Time vs. Batch Size). While the paper omits the partition function (Z) in Equation (3) to accelerate inference, this compromises probabilistic interpretability.
>
> The reviewer’s observation is accurate. Nonetheless, we kindly highlight that, even with the additional overhead, our approach introduces an additional computational overhead of only 2ms/image.

---

### Decision · Program_Chairs · 2025-05-01

**Decision:**

Accept (poster)

**Comment:**

This paper uses Markov Logic Networks (MLNs), integrating probabilistic reasoning with logical constraints, to realise improved structure and interpretability in out-of-distribution (OOD) detection. The reviewers found this paper (1) proposes novel approaches, (2) the experiments are solid and comprehsive, etc. which I mostly agree with. The reviewers also raised concerns, such as that overhead could incease, which are largely cleared. So, I recommend to accet this paper.